# Sepsis under pressure, intraoperative surgical site infection prevention practices among nurses in emergency surgical settings: A qualitative study

Racheal Nyarko[1¤], Eric Tornu[1], Gwendolyn Patience Mensah[2]*

1 Department of Adult Health Nursing, School of Nursing and Midwifery, College of Health Sciences, University of Ghana, Legon, Accra, Ghana, 2 Department of Maternal and Child Health, School of Nursing and Midwifery, College of Health Sciences, University of Ghana, Legon, Accra, Ghana

¤ Current address: Seventh Day Adventist (SDA) Nursing and Midwifery Training College, Kwadaso – Kumasi, Ghana
* gpmensah@ug.edu.gh

## Abstract

### Background

Surgical site infections are among the most common and preventable healthcare-associated infections globally. Nurses play a pivotal role in implementing preventive measures for surgical site infections. This study examined intraoperative surgical site infection prevention measures and their outcomes among nurses managing emergency surgeries at a tertiary hospital in Ghana.

### Methods

A qualitative exploratory descriptive design was employed. Nineteen nurses with at least three years of experience in the operating theatre were purposively selected from the surgical directorate of a teaching hospital in the Ashanti region of Ghana. Data were collected through semi-structured face-to-face individual interviews and analyzed thematically using Braun and Clarke's approach.

### Findings

The two themes and eight subthemes generated revealed that the structures influencing surgical site infection prevention encompassed the physical layout of the theatre, availability of equipment and materials, monitoring and supervision, institutional protocols, communication and teamwork and preoperative skin preparation. Furthermore, the outcomes of nursing interventions included improved wound healing and enhanced job satisfaction. However, resource limitations and inconsistent supervision posed barriers to optimal care. Nurses expressed fulfilment when wound healing progressed well, while infections led to emotional distress and increased workload.

**Data availability statement:** All relevant data are within the paper and its Supporting information files.

**Funding:** The author(s) received no specific funding for this work.

**Competing interests:** The authors have declared that no competing interests exist.

**Abbreviations:** ASA: American Society of Anaesthesiologists; CHG: Chlorhexidine; CSSDs: Central Sterilization Supply Departments; LMIC's: Low Middle -Income Countries; LTCS: Low Transverse Caesarean Section; IPAC: Isopropyl Alcohol; NHS: National Health Service; MDT: Multidisciplinary Team; QOL: Quality of Life; SSI's: Surgical Site Infections; THIN: The Health Improvement Networks; WHO: World Health Organization.

## Conclusion

Intraoperative infection prevention practices, when supported by strong institutional structures can significantly reduce surgical site infections and enhance patient outcomes. Investing in nurse-led infection control capacity, resource availability, and supportive supervision is essential for improving surgical safety in emergency contexts.

---

## Introduction

Surgical site infections (SSIs) continue to pose a significant and persistent challenge in modern healthcare, representing one of the most common postoperative complications worldwide [1] SSIs are infections that occur at or near a surgical incision within 30 days of an operative procedure, or within one year if an implant is placed. These infections are associated with substantial morbidity, mortality, prolonged hospital stays and increased healthcare costs [2–4].

Despite advancements in surgical techniques, antibiotic prophylaxis, and infection prevention protocols, SSI rates vary widely between 2.5% and 41.9% across healthcare settings globally, with an even higher burden observed in low and middle-income countries [5–7].

In high-income settings, SSIs account for approximately 20% of all hospital-acquired infections, but in low and middle-income countries, such as Ghana, they represent up to 40% of such infections [8–10] Emergency surgeries, characterised by their urgency, high contamination rates and limited time for patient optimisation, are particularly prone to SSIs [11,12]. Factors such as compromised patient conditions, prolonged surgical duration, and environmental limitations within operating theatres contribute to the heightened risk associated with these procedures. Postoperative infections following emergency surgeries significantly delay wound healing, escalate the need for antibiotics, increase healthcare costs, and adversely affect patients' quality of life [13,14].

The intraoperative phase is critically important for SSI prevention, representing a window where adherence to meticulous sterile techniques, timely antibiotic administration, appropriate surgical site preparation, and effective team communication can profoundly influence patient outcomes [15,16]. In this phase, nurses play an essential role alongside surgeons and anaesthetist by ensuring that aseptic protocols are strictly followed, environmental contamination is minimized, and patient care is optimised throughout the procedure [17,18]. However, in resource-limited settings like Ghana, barriers such as equipment shortages, staff constraints, and lapses in infection control practices often compromise intraoperative preventive efforts [19,20].

Previous studies in Ghana's tertiary hospitals have highlighted unfavourable ward and theatre environments, inadequate adherence to aseptic protocols, and limited patient preparation as factors contributing to the high SSI rates observed [21–23]. Due to sociocultural factors, these patients may not express their negative experiences for timely intervention [24]. Despite the recognised importance of

intraoperative preventive measures, there is a paucity of context-specific evidence exploring nurses' intraoperative SSI prevention practices in emergency settings and their outcomes. Understanding nurses' SSI prevention measures and outcomes can inform the design of effective SSI prevention and management interventions, strengthen clinical protocols, and ultimately improve the quality of surgical care. Therefore, this study aimed to explore the intraoperative SSI prevention measures and their outcomes among nurses managing emergency surgeries at a tertiary hospital in Ghana. The insights generated can address a critical knowledge gap and contribute to evidence-based improvements in surgical infection prevention within resource-constrained healthcare settings.

## Materials and methods

### Study design

This study adopted a qualitative exploratory descriptive design, which was appropriate for gaining an in-depth understanding of nurses' intraoperative preventive practices against SSIs and their perceived outcomes [25].

### Study setting

The study was conducted in the surgical wards and main theatre of the Directorate of Surgery at a public tertiary hospital in Kumasi, Ghana. The 1,200-bed referral hospital serves as a major referral health facility for Ghana's northern and middle sectors. The hospital's surgical directorate manages a high volume of emergency surgeries, making it an appropriate site for investigating intraoperative infection prevention practices.

### Study population

The study population comprised professional nurses working in the surgical wards and main theatre of the hospital. There are approximately 200 nurses within the surgical directorate. Participants were selected based on their experience and direct involvement in the care of patients undergoing emergency surgeries. Specifically, the study included nurses who had worked in the surgical wards or main theatre for at least three years, ensuring that participants had sufficient exposure and insights into SSI prevention practices. Nurses with less than three years of experience as surgical nurses were excluded from the study.

### Sampling technique and sample size

A purposive sampling technique was used to recruit nineteen [19] surgical nurses with rich and relevant experience managing emergency surgeries. The final sample size was guided by information power, which indicates that the more relevant information a sample holds, the fewer participants are needed [26]. However, data collection ended when two additional interviews yielded redundant beyond the point where participants' feedback became repetitive and additional interviews were not yielding new insights to augment the themes and subthemes. The participants included nurses from various units (main operating theatre, recovery ward, surgical ward) with varying years of experience (ranging from 3 to 15years), sexes (4 males and 15 females) and ranks (staff nurse, nursing officer, senior nurse, principal nursing officer and deputy chief nursing officer) to ensure diversity in perspectives.

### Researcher's positionality and reflexivity

The primary researcher is a registered nurse with academic training in nursing and prior clinical experience, but had no supervisory or clinical relationship with the participants. This outsider-insider perspective helped establish rapport while minimising potential power imbalances. The research team employed reflexive journals to document personal assumptions, methodological decisions, and reflections on the interview process. Reflexive memos were reviewed during analysis to minimise bias and enhance interpretive depth.

The researcher-maintained neutrality during interviews and analysis to ensure that participants' voices remained central.

## Data collection

Data were collected through individual face-to-face, semi-structured interviews using an interview guide informed by existing literature and the Donabedian model of healthcare quality [27,28]. The interview guide was pilot-tested with three nurses outside the study setting to ensure clarity and relevance. Interviews were conducted in English, lasted approximately 45 minutes each, and were audio-recorded with the participant's consent. Field notes were taken to capture non-verbal cues and contextual observations. Audio recordings were transcribed verbatim, anonymised, and securely stored on a password-protected device accessible only to the research team. Unique codes were assigned to transcripts to preserve participant confidentiality. Data Analysis Reflexive thematic analysis was conducted using Braun and Clarke's approach: (1) familiarisation with data, (2) generating initial codes, (3) theme generation, (4) refinement and (5) reporting (Braun & Clarke, 2020). Firstly, the researcher listened to the recorded interviews and transcribed them verbatim. Data was read and re-read multiple times to get a sense of the overall content and to identify any initial patterns while taking notes. Initial codes were generated inductively by two independent researchers who coded transcripts line-by-line. The two coding sets were compared, discrepancies were resolved through discussion, and a unified coding framework was developed.

Next, codes were organised into potential categories and later into broader, overarching themes. Themes were reviewed against the dataset to ensure coherence, internal consistency, and alignment with participant narratives. The research team refined theme definitions and constructed analytic summaries supported by representative quotations from participants.

Throughout analysis, reflexive memos were used to document analytic decisions, emerging interpretations, and potential biases. An audit trail was maintained, including coding frameworks, thematic maps, and meeting notes.

The themes were reviewed to ensure they accurately reflect the data and are relevant to the research purpose. Themes were used to create a narrative that tells the story of the data and addresses the research question. The analysis was conducted by researchers RN & GPM. We did inter-rater coding where researchers did initial coding independently. We then compared the codes, discussed disagreements and developed a coding frame to help with coding all the other transcripts.

Codes were derived inductively from the data. Themes were interpreted in relation to the study objectives. The recruitment of the study participants started on the 7th of May 2023 and ended on the 23rd of July 2023.

## Peer debriefing

Peer debriefing was incorporated throughout the study to strengthen the credibility and interpretive rigour of the findings. The core research team met every two weeks to discuss emerging codes, challenge assumptions, and critically examine alternative interpretations. An external qualitative expert, who was not involved in data collection, participated in two debriefing sessions to review the analytic decisions, question potential researcher bias, and confirm the coherence of themes. Notes from each peer-debriefing session were documented and used to refine the coding framework and ensure analytic transparency.

## Ethics approval and consent to participate

Ethical approval for the study was obtained from the Institutional Review Board of Komfo Anokye Teaching Hospital (KATH IRB/AP/040/23), a tertiary hospital in Ghana. Administrative clearance was also obtained from the tertiary health facility within which participants were sampled. The study followed the guidelines of the Declaration of Helsinki. Written

informed consent was obtained from all participants prior to data collection, following detailed explanations about the study's purpose, procedures, confidentiality assurances, and voluntary participation. Participants were assured of their right to withdraw at any stage without penalty. Anonymity was preserved by using codes instead of names, and audio recordings and transcripts were securely stored with restricted access.

### Trustworthiness

To ensure trustworthiness, the study adhered to the criteria outlined by Lincoln and Guba (1985). Credibility was established through prolonged engagement with participants, member checking, and iterative questioning techniques during interviews. Transferability was supported by providing rich, thick descriptions of the context, participants, and procedures, enabling readers to assess applicability to other settings. Collectively, these strategies ensured that the findings were robust, authentic, and accurately reflected the participants' true experiences. Dependability was maintained by creating a comprehensive audit trail detailing the research processes and decisions. Confirmability was ensured by maintaining reflexive notes and conducting peer debriefing sessions to minimise researcher bias. The researcher ensured this by transcribing the information from the participants immediately to prevent misinterpretation in meaning. The researcher also developed an adequate audit trail which contained field notes, audio-recordings, interview transcripts and documents on emerging themes and categories, notes from the member check, personal notes from the field and interpretation including draft from the final report. Member-checking was conducted with eight participants to validate emerging interpretations. This audit enquiry ensured confirmability of the data. The researcher also, examined the beliefs, values and experiences of participants and did self-reflection on her beliefs, values and experiences in the same manner. The researcher's beliefs and values were made clear and taken into account so as to help the researcher avoid biases.

## Findings

### Participants' characteristics

Nineteen (19) nurses, aged 27–43 years and comprising 15 females, participated in the study. Thirteen (13) participants were stationed on the surgical wards, while six (6) worked in the main theatre. Participants had between 3 and 21 years of professional nursing experience, and 14 held a Bachelor of Science in nursing degree. Table 1 details the participants' characteristics.

### Themes and subthemes

Table 2 presents the two themes and eight subthemes generated from the data.

   **Theme 1: Structures influencing surgical site infection prevention.** This theme describes participants' perspectives on how organisational, environmental, and procedural factors influenced nurses' intraoperative SSI prevention practices. The six sub-themes generated under this theme included (1) physical layout of the theatre, (2) equipment, materials, and resources, (3) monitoring and supervision, (4) organisational policies and protocols, (5) communication and team approach, and (6) skin preparation for surgery.

   Physical layout of the theatre: The participants identified the physical layout of the operating theatre as crucial for infection control. They emphasised the importance of zoning (unrestricted, semi-restricted, and restricted areas) and the use of appropriate personal protective equipment within the zones of the theatre.

> "We have a restricted, semi-restricted and unrestricted area in the theatre... Nurses need to change in the changing room to wear scrubs, boot, and hair cover before moving to the semi-restricted area. The restricted area is where the sterile cases and the operations are performed where you need to change completely from your head to toe." (P7)

**Table 1. Participants' characteristics.**

| Participants | Age (years) | Sex | Surgical Unit | Educational level | Years of Experience |
|---|---|---|---|---|---|
| P1 | 37 | Female | Main theatre | Degree | 10 |
| P2 | 40 | Female | Ward (B2) | Degree | 7 |
| P3 | 33 | Female | Ward (B2) | Degree | 9 |
| P4 | 30 | Female | Main theatre | Diploma | 4 |
| P5 | 37 | Female | Ward (B3) | Degree | 13 |
| P6 | 31 | Female | Ward (B1) | Diploma | 5 |
| P7 | 34 | Male | Ward (B1) | Degree | 9 |
| P8 | 27 | Female | Main theatre | Degree | 3 |
| P9 | 32 | Female | Main theatre | Degree | 10 |
| P10 | 39 | Female | Main theatre | Degree | 10 |
| P11 | 33 | Female | Ward (B3) | Degree | 6 |
| P12 | 43 | Female | Main theatre | Masters | 21 |
| P13 | 29 | Male | Ward (B2) | Diploma | 3 |
| P14 | 32 | Male | Ward (B2) | Degree | 6 |
| P15 | 35 | Female | Ward (C4) | Degree | 10 |
| P16 | 38 | Female | Ward (C3) | Masters | 11 |
| P17 | 33 | Male | Ward (B3) | Degree | 6 |
| P18 | 34 | Female | Ward (C3) | Degree | 6 |
| P19 | 40 | Female | Ward (C3) | Degree | 15 |

Maintaining a hygienic environment was emphasised. Participants noted that daily and scheduled high-dusting routines, combined with strict cleaning protocols using appropriate disinfectants, contributed to infection control.

"We do high dusting unlike to normal dusting... We clean our instruments with decontaminating agents such as Ultra –P instead of bleach" (P1)

Equipment, materials and resources: All participants noted that the sufficient availability of surgical materials, such as sterile gloves, antiseptics, and instruments, was essential for preventing infections. However, many reported inconsistent supplies, which often necessitated improvisation.

"Sometimes we run out of stock, especially with the disinfecting solutions. Sometimes we run out of solutions like methylated spirit and others during procedures." (P3)

"During the COVID era, it was difficult to get gowns for work, so the local factories had to produce the gowns." (P2)

Participants emphasised that while dressing materials were prioritised, other critical supplies occasionally became limited, requiring improvisation.

Monitoring and supervision: Several participants emphasised the significance of monitoring and supervision in ensuring adherence to SSI prevention protocols. They noted that both formal oversight by those in charge and departmental heads, as well as individual accountability among nurses, contributed to safe surgical practices and infection prevention.

"Every shift has an in-charge to ensure that the right thing is being done." (P15).

**Table 2. Themes and sub-themes.**

| Theme | Subtheme | Codes |
|---|---|---|
| 1. Structures influencing surgical site infection prevention | 1.1 Physical layout of the theatre | • Theatre zoning: restricted, semi-restricted, unrestricted<br>• Personal protective equipment expectations<br>• High-dusting vs routine cleaning<br>• Environmental hygiene and sterile field maintenance |
| | 1.2 Equipment, materials and resources | • Availability of sterile supplies<br>• Shortages of disinfectants and gowns<br>• Improvisation due to supply gaps<br>• COVID-era adaptations |
| | 1.3 Monitoring and supervision | • Shift in-charge oversight<br>• Departmental supervision structures<br>• Internal motivation and self-regulation<br>• Lapses when supervision is absent |
| | 1.4 Organisational policies and protocols | • Institutional guidelines for infection prevention<br>• Protocol posters as reminders<br>• Hand hygiene, patient preparation, and environmental cleaning procedures<br>• Protocols embedded in daily workflow |
| | 1.5 Communication and team approach | • Multidisciplinary communication platforms<br>• Surgeons' involvement via messaging channels<br>• Collaborative rounds between nurses and doctors<br>• Team responsiveness for emergencies |
| | 1.6 Skin preparation for surgery | • Hair removal techniques<br>• Antiseptics used: Savlon, spirit, povidone-iodine<br>• Variations between elective and emergency procedures<br>• Ensuring minimum preparation under time pressure |
| 2. Outcome of nurses' intervention in surgical site infection prevention | 2.1 Wound healing | • Healing by primary intention<br>• Occasional postoperative wound infections<br>• Secondary suturing and prolonged dressing<br>• Financial and system burden (bed occupancy, cost, infection risk) |
| | 2.2 Job satisfaction | • Sense of achievement with good outcomes<br>• Emotional relief when wounds heal<br>• Stress, frustration, and self-doubt from infected wounds<br>• Reflective practice and desire for improvement |

"It starts with you…as a nurse with knowledge of infection risks, I'm motivated to do the right thing. Then the supervisor also comes in to see that you are doing the right thing." (P7).

However, some participants noted that supervision was not always consistent, particularly during off-peak hours, resulting in lapses in adherence to infection prevention protocols.

"When the (nurse) in-charge is not around, especially during afternoon and night shifts, some staff tend to do things the way they want, and since there is no one monitoring what is going on, they do whatever they like." (P14)

These findings suggest that while structured oversight strengthens protocol adherence, sustainable infection control also depends on cultivating a culture of internal motivation and mutual accountability among nurses.

Organisational policies and protocols: Participants widely acknowledged that organisational protocols played a crucial role in guiding their infection prevention practices throughout the surgical care continuum. Several participants referred to institutional policies as essential frameworks that structured their daily routines, from pre-operative preparation to post-operative care. To improve accessibility, participants observed that protocols were prominently displayed within the wards and theatre units to act as ongoing references for both new and experienced staff.

"There are numerous (infection prevention) protocols. We have protocols in place for personnel such as theatre staff, the nurses and the patients." (P1)

"We have posters around. We have wound dressing protocol and hand washing protocol posters so that the average person who doesn't know can look at them and follow them." (P18)

Participants explained their commitment to established protocols concerning hand hygiene, patient preparation, and environmental cleanliness, going beyond simple documentation. These protocols outlined procedures, such as shaving patients on the operating table to minimise pre-surgical contamination, decontaminating instruments daily, and enforcing strict staff hygiene.

"We perform hand washing, wear nose mask and protective cloth as in gown" (P2)

These responses reflect how institutional protocols were not only documented but actively embedded in the practice culture, reinforcing a standard of safety and accountability in infection prevention.

Communication and team approach: Effective communication and teamwork among nurses and interdisciplinary healthcare providers were consistently identified as essential to preventing SSIs. Participants highlighted how timely information sharing, collaborative decision-making, and open communication channels across surgical teams facilitated the early detection and response to wound-related complications or infections.

"We have a platform that consists of the surgeons and the nurses. So, in the absence of the surgeons, you put it on the platform. Sometimes, you can get an emergency situation, such as a burst abdomen (dehiscence) from the ward, we report it there, and the surgeons respond quickly" (P2)

Participants also noted that consistent communication between nurses and physicians during ward rounds and post-operative care allowed for rapid intervention and treatment adjustments.

"When there are signs of wound infections say, greenish discharges, offensive odour, we mostly discuss with the doctors, and they will either add some antibiotics or ask that we change the dressing routinely. The nursing staff also make sure that the changes made are implemented," (P16).

In addition, the presence and accessibility of multidisciplinary teams, including consultants, pharmacists, and physiotherapists, was seen as a key strength in wound management.

"You'll find all the disciplinary teams coming in for ward rounds. So, you find a consultant, the medical officers, the residents, house officers, dieticians, physiotherapists, pharmacists—everybody is involved. Even when they don't come around, they are just a call away. We have their phone numbers, so we just call them when needed." (P11)

Skin preparation for surgery: Participants widely emphasised the significance of pre-operative skin preparation, particularly shaving and cleansing the surgical site, as a fundamental step in preventing SSIs. Several participants described standard practices such as removing hair around the incision site and cleaning the area with antiseptics like Savlon, methylated spirit, and povidone-iodine before surgery.

"We do shave them, especially the hair on the incisional area from the mid epigastrium to the symphysis pubis, yes, we do shave them. Then we clean before sending them to the theatre" (P5)

Participants also highlighted the impact of emergency contexts, which often limited the time available for thorough pre-operative preparation. Despite these constraints, nurses reported making every effort to ensure at least the basic cleaning of the site prior to surgery.

"Usually during emergency cases, we are not able to go through the normal preparations we do but in as much as we are in a hurry, we are able to at least prepare the surgical site with savlon, methylated spirit and povidone-iodine. The patient is draped before taking the patient to the theatre" (P14)

These responses reflect a consistent awareness of best practices in skin preparation and a strong commitment to infection prevention, even under the time pressures inherent in emergency surgery.

**Theme 2: Outcome of the nurses' intervention in surgical site infection prevention.** Participants described the outcomes associated with their SSI prevention interventions. The two sub-themes identified under this theme included (1) wound healing and (2) job satisfaction.

Wound healing: Participants identified wound healing as a key indicator of the effectiveness of their SSI prevention measures. Many participants noted that when established protocols were adhered to, such as aseptic dressing, timely administration of medication, and patient education, surgical wounds generally healed by primary intention.

"Previously, we saw many wounds showing signs of infection but because of the structures put in place and how the nurses provide care, most of our wounds heal well without infection". (P9)

Despite these positive outcomes, participants also noted that infections occasionally occurred, particularly in complex cases, leading to extended dressings, secondary suturing, and longer hospital stays for postoperative patients.

"We had a patient who was here because his wound was infected, and he had to stay in the hospital for almost two weeks before he was discharged after secondary suturing." (P15)

Several nurses emphasised that poor wound healing had wider implications, including a greater financial burden for patients, prolonged hospital stays and increased pressure on hospital resources.

"It brings additional cost to the patient, the nurses, and the hospital's management. At times, when the patient has to go for a secondary suturing, it adds to the patient's cost as well as increases bed occupancy, making it difficult to admit new patients. It also prolongs their stay in the hospital and ultimately leads to nosocomial infections." (P7)

These insights illustrate the tangible impact of nursing interventions on wound outcomes while highlighting the clinical and systemic implications of delayed healing.

Job satisfaction: Participants described a strong emotional connection between patients' outcomes and their own professional fulfilment. Many expressed that successful wound healing saved them from stress and brought them satisfaction, relief and a sense of accomplishment.

"When surgery is successfully done and the patient is discharged, you feel that you have done what you are supposed to do". (P3)

"It is stress-saving. It saves you from the stress of dressing wound continuously and trying a lot of solutions to help the wound heal." (P2)

Conversely, when postoperative patients' wounds became infected, the participants frequently experienced disappointment, self-doubt, and emotional stress, prompting them to reflect on potential lapses in care or infection prevention practices.

"When the wound is infected, you become sad and feel like you did not do something right. You start reflecting and asking, 'Is it because during the surgery something went wrong?'… It is frustrating" (P2)

These responses underscore how nurses' job satisfaction is closely tied to patient recovery and how adverse outcomes can impact their emotional well-being and sense of professional efficacy.

## Discussion

This study explored intraoperative SSI prevention practices among nurses managing emergency surgeries in a tertiary facility in Ghana. The findings demonstrate that nurses apply both institutional protocols and personal initiative to reduce infection risks, even in the face of systemic constraints common to resource-limited settings. Participants emphasised the value of clear theatre zoning and daily cleaning routines in maintaining sterile environments. These structural supports echo global infection control standards [29]. However, recurring shortages of essential supplies such as sterile gloves, antiseptics, and protective gowns often challenged these efforts. This aligns with evidence from other low-resource settings showing that stock-outs weaken compliance, even when protocols are known [30]. Despite these constraints, visible infection control protocols and posters on wound care, hand hygiene, and instrument decontamination helped sustain standard practices. These tools provided just-in-time reinforcement and were particularly useful for guiding less experienced staff. Formal supervision by nurse managers and in-charges was regarded as essential for reinforcing best practices. Participants observed greater compliance when supervisors were present. However, limited oversight during night shifts occasionally led to reduced adherence. These observations support findings that routine monitoring enhances infection control performance [31]. Notably, many nurses described internal motivation and peer accountability as complementary drivers of safe practice, implying that professional culture is a valuable asset in infection control. Effective communication and multidisciplinary collaboration were crucial for SSI prevention. Nurses highlighted the timely reporting of wound concerns through shared platforms and the importance of supportive teamwork during ward rounds. These mechanisms facilitated rapid responses to infection risks and align with the literature, which indicates that integrated care enhances surgical outcomes [18].

Emergency cases often limit the time available for full pre-operative preparation. Nonetheless, the nurses consistently reported performing essential cleansing with antiseptics such as Savlon, spirit, and povidone-iodine. This highlights nurses' adaptability and the need for flexible infection control strategies in emergency contexts [32]. Wound healing was widely cited as a visible outcome of effective nursing care. Nurses connected proper wound dressing, patient education, and aseptic technique to healing by primary intention. This reflects previous studies demonstrating that nurse-led interventions reduce SSI risk and facilitate recovery [33]. Conversely, infections resulted in prolonged hospital stays, secondary suturing, and increased patient costs, burdens also reported in other low-and middle-income settings [34–36]. Emotional responses were closely linked to patient outcomes. Nurses expressed satisfaction and pride when wounds healed well but felt frustration and guilt when infections occurred. These findings support qualitative research indicating that nurses' morale is deeply affected by clinical results [37]. The findings of this study highlight several priorities, such as ensuring an uninterrupted supply of infection control materials, extending effective supervision across all shifts, embedding practical, visible protocols within clinical spaces and recognising the emotional investment of nurses as a quality factor in surgical care.

## Conclusion

This study offers important insights into intraoperative SSI prevention from the perspective of nurses managing emergency surgeries in a resource-constrained tertiary hospital in Ghana. Despite systemic limitations, such as equipment

shortages and time pressures, nurses demonstrated a strong commitment to infection control through adherence to protocols, personal accountability, and effective team communication. Positive outcomes, such as wound healing and job satisfaction, were observed when these practices were implemented consistently. Conversely, lapses in supervision or protocol enforcement were associated with infection-related complications and extended hospital stays. The findings affirm that structural support, material resources, and a culture of professional accountability are critical to improving surgical outcomes in emergency contexts.

## Acknowledgments

The authors are grateful to the leadership and staff of the surgical directorate at Komfo Anokye Teaching Hospital for their unwavering support and cooperation throughout the course of this study. Our gratitude also goes to the participants who generously shared their perspectives on nursing care with us.

## Author contributions

**Conceptualization:** Racheal Nyarko, Eric Tornu, Gwendolyn Patience Mensah.

**Data curation:** Racheal Nyarko, Eric Tornu, Gwendolyn Patience Mensah.

**Formal analysis:** Racheal Nyarko, Eric Tornu, Gwendolyn Patience Mensah.

**Methodology:** Racheal Nyarko, Eric Tornu, Gwendolyn Patience Mensah.

**Supervision:** Eric Tornu, Gwendolyn Patience Mensah.

**Validation:** Racheal Nyarko, Eric Tornu, Gwendolyn Patience Mensah.

**Visualization:** Racheal Nyarko, Eric Tornu, Gwendolyn Patience Mensah.

**Writing – original draft:** Racheal Nyarko.

**Writing – review & editing:** Racheal Nyarko, Eric Tornu, Gwendolyn Patience Mensah.

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
