## [Decision Letter · Decision Letter 0]

26 Oct 2025

Dear Dr. Mensah,

Thank you for submitting your manuscript to PLOS One. Firstly, we would like to apologize for the delay in processing your manuscript. It has been exceptionally difficult to secure reviewers to evaluate your study. We have now received one completed review, which is available below. The reviewer has raised significant scientific concerns about the study that need to be addressed in a revision.

Please note that we have only been able to secure a single reviewer to assess your manuscript. We are issuing a decision on your manuscript at this point to prevent further delays in the evaluation of your manuscript. Please be aware that the editor who handles your revised manuscript might find it necessary to invite additional reviewers to assess this work once the revised manuscript is submitted. However, we will aim to proceed on the basis of this single review if possible. 

We look forward to receiving your revised manuscript.

Kind regards,

Miquel Vall-llosera Camps

Senior Staff Editor

PLOS ONE

Journal Requirements:

Reviewers' comments:

Reviewer's Responses to Questions

**Comments to the Author**

1. Is the manuscript technically sound, and do the data support the conclusions?

Reviewer #1: Partly

2. Has the statistical analysis been performed appropriately and rigorously?

Reviewer #1: No

3. Have the authors made all data underlying the findings in their manuscript fully available?

Reviewer #1: No

4. Is the manuscript presented in an intelligible fashion and written in standard English?

Reviewer #1: Yes

Reviewer #1: 1. The research design aligns with the study aim of understanding intraoperative surgical site infection (SSI) prevention practices among nurses in a tertiary hospital. The use of semi-structured interviews and thematic analysis is suitable for the exploratory nature of the work.

The data collection procedures are adequately described, and the inclusion of direct participant quotations supports the authenticity of the findings. However, the analytic process lacks sufficient transparency. There is minimal description of the coding framework, how the codes were developed, refined and integrated into the themes. Although NVivo software is mentioned, its role in data management or theme generation is not clearly explained.

The findings and conclusions are largely consistent, with the data supporting the main claims about infection prevention practices, supervision, and resource challenges. However there is overgeneralization —for example, “multidisciplinary collaboration significantly reduces SSIs” that extends beyond what qualitative data can demonstrate.

There should be explicit documentation of ethics approval (e.g., institutional review board reference number).

Overall, the manuscript requires greater methodological transparency and moderation of inferential claims to fully substantiate its conclusions.

2. The manuscript provides only a superficial description of the analytic process: It lists Braun & Clarke’s phases but does not show how themes were derived from codes. There are no sample codes, codebooks, or excerpts to illustrate analytic progression. The process of reflexivity and peer debriefing is mentioned but not demonstrated with examples. while the method is suitable, the reporting of its application lacks depth and transparency.

3. No details on inter-coder reliability or peer validation (e.g., whether another researcher cross-checked coding).

No clear evidence of data triangulation or how saturation was confirmed. There is Over-reliance on researcher interpretation without documented audit examples.

4. The manuscript lacks frequency indicators for reporting prevalence without quantifying. There is no comparative analysis (e.g., by role, experience, or theatre vs ward) explored, which could have added analytic richness.

5. The manuscript does not include a Data Availability Statement, nor does it indicate where the underlying qualitative data (e.g., anonymized transcripts, coding framework) are deposited.

6. The manuscript is presented in an intelligible and well-structured manner overall. The English is clear though minor grammatical and stylistic issues should be addressed g. Some sentences are overly long or repetitive,

**Do you want your identity to be public for this peer review?** For information about this choice, including consent withdrawal, please see our Privacy Policy

Reviewer #1: No

---

## [Author Response · Author response to Decision Letter 1]

7 Dec 2025

The manuscript have been revised to meet PLOS ONE's style requirements.

The ethics statements have been moved to the methods section of the manuscript as recommended. (Page 9)

4. The relevant anonnymised data have been provided in the manuscript. (Pages 10-21)

We will update your Data Availability statement on your behalf to reflect the information you provide4.

Thank you

1. The research design aligns with the study aim of understanding intraoperative surgical site infection (SSI) prevention practices among nurses in a tertiary hospital. The use of semi-structured interviews and thematic analysis is suitable for the exploratory nature of the work.

The authors are grateful for the feedback.

2. The data collection procedures are adequately described, and the inclusion of direct participant quotations supports the authenticity of the findings. However, the analytic process lacks sufficient transparency. There is minimal description of the coding framework, how the codes were developed, refined and integrated into the themes. Although NVivo software is mentioned, its role in data management or theme generation is not clearly explained.

The analytical processes used have been provided as recommended.

The table of themes, sub-themes and codes have been provided to show how codes were developed, refined and integrated into the final themes. (Pages 7 & 8)

3. The findings and conclusions are largely consistent, with the data supporting the main claims about infection prevention practices, supervision, and resource challenges. However there is overgeneralization —for example, “multidisciplinary collaboration significantly reduces SSIs” that extends beyond what qualitative data can demonstrate.

We are grateful for this feedback. The overgeneralizing statements have been revised and limited to the qualitative data in the study (page 3)

4.There should be explicit documentation of ethics approval (e.g., institutional review board reference number).

Overall, the manuscript requires greater methodological transparency and moderation of inferential claims to fully substantiate its conclusions.

The institutional review board reference number was provided in the first document under Declarations section (Ethical Approval and Consent to participate). It has now been moved to the methods section. (Page 9)

5. The manuscript provides only a superficial description of the analytic process: It lists Braun & Clarke’s phases but does not show how themes were derived from codes. There are no sample codes, codebooks, or excerpts to illustrate analytic progression. The process of reflexivity and peer debriefing is mentioned but not demonstrated with examples. while the method is suitable, the reporting of its application lacks depth and transparency.

We appreciate this feedback. A more detailed description of how themes (and sub-themes) were derived from codes have been provided as recommended. The process of reflexivity and peer debriefing has also been detailed. (Pages 7, 8 & 9,

12,13 & 14).

6. No details on inter-coder reliability or peer validation (e.g., whether another researcher cross-checked coding).

No clear evidence of data triangulation or how saturation was confirmed. There is Over-reliance on researcher interpretation without documented audit examples.

Additional details have been provided regarding peer validation, particularly how the other researchers cross-checked the preliminary coding.We have added that saturation was confirmed by conducting 2 additional interviews beyond the point where participants’ feedback became repetitive, additional interviews were not yielding new insights to add up to the themes and sub-themes. (6, 8 & 9)

7. The manuscript lacks frequency indicators for reporting prevalence without quantifying. There is no comparative analysis (e.g., by role, experience, or theatre vs ward) explored, which could have added analytic richness.

We are grateful for this feedback. As recommended, we will explore these comparative analysis in follow-up studies to enhance the analytic richness.

8. The manuscript does not include a Data Availability Statement, nor does it indicate where the underlying qualitative data (e.g., anonymized transcripts, coding framework) are deposited.

The Data Availability Statement was provided in the manuscript (page 20) under the declarations section (Availability of data and materials). (Page 25)

9. The manuscript is presented in an intelligible and well-structured manner overall. The English is clear though minor grammatical and stylistic issues should be addressed. Some sentences are overly long or repetitiveWe are grateful for the feedback.

We have reduced the overly long or repetitive sentences as recommended. Grammatical errors have been corrected. (Page 23)

---

## [Decision Letter · Decision Letter 1]

28 Dec 2025

Sepsis under pressure, intraoperative surgical site infection prevention practices among nurses in emergency surgical settings: A qualitative study

PONE-D-25-42877R1

Dear Dr. Mensah,

We’re pleased to inform you that your manuscript has been judged scientifically suitable for publication and will be formally accepted for publication once it meets all outstanding technical requirements.

Kind regards,

Nan Jiang

Academic Editor

PLOS One

Reviewers' comments:

Reviewer's Responses to Questions

**Comments to the Author**

Reviewer #1: All comments have been addressed

2. Is the manuscript technically sound, and do the data support the conclusions?

Reviewer #1: Yes

3. Has the statistical analysis been performed appropriately and rigorously?

Reviewer #1: Yes

4. Have the authors made all data underlying the findings in their manuscript fully available?

Reviewer #1: Yes

5. Is the manuscript presented in an intelligible fashion and written in standard English?

Reviewer #1: Yes

Reviewer #1: All the comments previously raised have been addressed by the authors with improvements in the methodology thereby strengthening the manuscript

**Do you want your identity to be public for this peer review?** For information about this choice, including consent withdrawal, please see our Privacy Policy

Reviewer #1: No

---

## [Editor Report · Acceptance letter]

PONE-D-25-42877R1

PLOS One

Dear Dr. Mensah,

I'm pleased to inform you that your manuscript has been deemed suitable for publication in PLOS One. Congratulations! Your manuscript is now being handed over to our production team.

Kind regards,

on behalf of

Dr. Nan Jiang

Academic Editor

PLOS One